# Midline and Mediolateral Episiotomy: Risk Assessment Based on Clinical Anatomy

**DOI:** 10.3390/diagnostics11020221

**Published:** 2021-02-02

**Authors:** Danielle K. Garner, Akash B. Patel, Jun Hung, Monica Castro, Tamar G. Segev, Jeffrey H. Plochocki, Margaret I. Hall

**Affiliations:** 1Arizona College of Osteopathic Medicine, Midwestern University, Glendale, AZ 85308, USA; dgarner74@midwestern.edu (D.K.G.); apatel63@midwestern.edu (A.B.P.); jhung62@midwestern.edu (J.H.); tgsegev@gmail.com (T.G.S.); 2College of Graduate Studies, Midwestern University, Glendale, AZ 85308, USA; mcastr@midwestern.edu; 3Department of Medical Education, College of Medicine, University of Central Florida, Orlando, FL 32827, USA

**Keywords:** bulbs of the vestibule, midline episiotomy, mediolateral episiotomy, perineal nerve

## Abstract

Episiotomy is the surgical incision of the vaginal orifice and perineum to ease the passage of an infant’s head while crowning during vaginal delivery. Although episiotomy remains one of the most frequently performed surgeries around the world, short- and long-term complications from the procedure are not uncommon. We performed midline and mediolateral episiotomies with the aim of correlating commonly diagnosed postepisiotomy complications with risk of injury to perineal neuromuscular and erectile structures. We performed 61 incisions on 47 female cadavers and dissected around the incision site. Dissections revealed that midline incisions did not bisect any major neuromuscular structures, although they did increase the risk of direct and indirect injury to the subcutaneous portion of the external anal sphincter. Mediolateral incisions posed greater risk of iatrogenic injury to ipsilateral nerve, muscle, erectile, and gland tissues. Clinician discretion is advised when weighing the potential risks to maternal perineal anatomy during vaginal delivery when episiotomy is indicated. If episiotomy is warranted, an understanding of perineal anatomy may benefit diagnosis of postsurgical complications.

## 1. Introduction

Episiotomy is the surgical incision of the vaginal orifice and perineum to ease the passage of an infant’s head while crowning during vaginal delivery. Episiotomy remains one of the most commonly performed surgeries around the world, although routine episiotomy has been on the decline since guidelines from multiple obstetric societies recommended against its use, citing insufficient evidence of its efficacy [1,2,3,4]. However, episiotomy remains an important part of the obstetrician’s toolkit (even in the United States) during emergencies of fetal distress in the presence of a tight maternal perineum, especially in the case of shoulder dystocia [5,6]. However, in the same time period that routine episiotomy has fallen out of favor, obstetric and sphincter injuries, termed OASIS, have been on the rise [7,8,9,10]. OASIS is a serious maternal health concern that is associated with maternal morbidities, including pelvic floor dysfunction, fecal and urinary incontinence, sexual dysfunction, and pelvic organ prolapse [11,12,13]. In response to the increase in perineal injury during delivery, the prevention and management of OASIS has been deemed a priority by the international obstetric community [3,14,15].

The relationship between OASIS and episiotomy is difficult to elucidate. Episiotomy has been reported to both mitigate the risk of OASIS and be a risk factor for OASIS [2,16,17,18]. These conflicting findings may be explained by variations in episiotomy incisions, with midline incisions posing greater risk of perineal tearing and mediolateral incisions resulting in relatively fewer complications, especially those associated with the development of fecal continence [19,20,21]. It has been reported that deep perineal tears occurred in 14.8% of vaginal births using midline episiotomy, compared to only 7.0% of births with mediolateral episiotomies in the same timespan [22]. However, consistency in the placement of mediolateral incisions among practitioners has been questioned, confounding attempts to precisely evaluate the risks of episiotomy by incision location, angle, and depth [23,24,25]. Consequently, the relationship between diagnoses of episiotomy-related maternal morbidities and surgical incision type remain unclear, as is the role of perineal anatomic variation in birth-related injuries.

In this study, we utilize human cadaveric dissection to quantify anatomic variation in the female perineum as it relates to common episiotomy approaches. Our aim is to provide clinicians with anatomic evidence that may be used in the decision-making process to weigh the risk of injury to perineal anatomy if episiotomy is indicated during vaginal childbirth. We further aim to improve diagnoses of postepisiotomy complications by correlating perineal anatomical variation with common negative outcomes. We focus on injury risk as it relates to the origin and orientation of common episiotomy incision locations. A better understanding of perineal anatomic variation in the incisive field of episiotomy may help mitigate the risk of OASIS and other birth-related injuries. We hypothesize that each contemporary episiotomy method endangers unique perineal anatomy, and here we denote the structures at risk for these different approaches.

## 2. Materials and Methods

This work was funded by Midwestern University. We obtained 47 donated female human cadavers from the National Body Donor Program, St. Louis, MO, USA, a portion of the cadavers utilized by Midwestern University in medical and health sciences gross anatomy courses. All specimens were embalmed via the internal jugular vein with six gallons of embalming fluid (3% formaldehyde, 4% phenol, 31% glycerin, and 31% water). After embalming, the cadaver was stored at room temperature for a minimum of one month before delivery to Midwestern University. All cadavers were treated in accordance with local and national laws and regulations. All tissues were observed at dissection after cadaver donation was complete, and no identifying information was known for any cadaver beyond age and cause of death. Therefore, we obtained a release from the Midwestern University IRB for cadaver use in this study. We excluded any cadavers with perineal pathologies or other perineal abnormalities. The presence of scarring from past episiotomy was also noted, but due to the privacy laws associated with cadaveric donation we did not have any specific information on the obstetrical history of the donors. However, the average age of the cadavers was 72.43 with a range of 52–99, indicating that the cadavers were likely postmenopausal.

A total of 61 dissections were completed, including episiotomy incisions through superficial tissue layers were made in the midline (*n* = 31) and in the mediolateral direction at angles below 15° (low, *n* = 9), between 15 and 44° (medium, *n* = 10), and 45° and above (high, *n* = 11) prior to complete dissection of the perineum (Figure 1). We performed the incisions using standard small, sharp dissection scissors with one blade of the scissors inside the vaginal orifice and the other directly opposite outside the orifice. The majority of cadavers received both a midline and one of the mediolateral angle variants to make fuller use of the donated sample. It is important to note that the angle of the incision made during crowning does not remain constant after delivery due to anatomic changes in the vaginal wall and surrounding area [26]. The angles used in our study mirror the normal range of episiotomy scarring angles based on postpartum observations [7,16,27]. By defining incision angles in this manner, we were able to ensure that the locations of our incisions on nonpregnant cadavers coincided with those performed during crowning.

Subsequent to incision, we measured the length of the episiotomy cuts using digital calipers to the nearest mm to ensure consistency. The incision field was then examined to identify structures transected during the procedure. We cleaned and observed nerve, muscle, erectile, and gland tissues in the incisive plane to assess injury risk. We next dissected away from the cut edges of the incision using blunt and standard dissection equipment, including scalpels, scissors, and probes where appropriate, following diagrams by Lappen [28]. We then traced the posterior labial nerves to their origin from the superficial perineal nerve, turned the cadavers prone, and followed the superficial and deep perineal nerves to their origin from the pudendal nerve. The locations of these anatomical structures were recorded for each individual in the sample and then aggregated to generate a “heat map”(Figure 5). The colors of the heat map reflect the risk of injury based on the number of structures exposed in the incisive field.

## 3. Results

Results of our dissections are described below, organized by episiotomy incision type, as seen in Figure 1a,b. Figure 2 displays cadaveric episiotomy incisions. Figure 3 shows the most common configuration of perineal anatomy in our sample, and Figure 4 shows one of the cadaveric dissections revealing relevant anatomy to Figure 3. Figure 5 aggregates the anatomic variation in our sample into a “heat map”, with risk of injury represented by a matrix of colors. Red indicates the greatest risk, with over 80% of cadavers having major nerve, muscle, erectile, or gland structures in the incisive plane. Blue indicates low risk, with fewer than 20% of cadavers having major structures in the incisive plane.

### 3.1. Midline Incision

In 100% of cadavers, the midline incision damaged the connective tissue of the perineal body and overlying skin (see Table 1). In 16.6% of cadavers, we identified gracile muscle fibers inserting onto the perineal body, while 83.4% had only connective tissue within the incision. The predominant collagen fiber direction of the perineal body coursed parallel to the midline incision in a sagittal orientation. In 14.3% of cadavers, the incision was of sufficient length to completely bisect the perineal body and extend to some muscle fibers of the subcutaneous portion of the external anal sphincter (EAS), causing damage. However, in these instances the superficial and deep portions of the EAS were left intact, either because the majority of fibers did not converge at the midline and instead were continuous with fibers of the bulbospongiosus in the parasagittal plane, or they were sufficiently deep to avoid direct insult by the scissor blades. In the remaining 85.7% of cadavers, all portions of the EAS were undamaged by the incision. The internal anal sphincter was not directly threatened in any cadaver, nor were any major neurovascular structures located in the vicinity of the incision.

### 3.2. High-Angle Mediolateral Incision

In 100% of cadavers, the bulbospongiosus muscle and bulb of the vestibule were located in the plane of the incision (see Table 1). Injury to these structures was depth-dependent. Incisions greater than the combined depth of the skin and superficial fascia, which was usually no more than a few millimeters, led to muscle and erectile tissue damage. However, this thickness may not be representative of the thickness during crowning, when these structures are stretched. In 40% of cadavers, there was damage to branches of the deep perineal nerve, branches of the superficial perineal nerve, and the main trunk of the superficial perineal nerve, all of which coursed primarily in the anteromedial direction within the incision field. Therefore, the terminal branches of the superficial perineal nerve were also affected, including the posterior labial nerves, which innervate the labia minora and most of the labia majora. No direct injury to the perineal body or EAS occurred.

### 3.3. Medium-Angle Mediolateral Incision

In 100% of cadavers, posterior labial nerves coursed through the incisive plane within the superficial fascia just below the dermis of the skin and were bisected in our dissections (see Table 1). In 80% of our sample, the bulbs of the vestibule and the bulbospongiosus muscle were within the incisive plane just deeper than the superficial fascia. In total, 40% of cadavers we incised exhibited damage to the greater vestibular (Bartholin’s) gland, which was consistently located in the superficial fascia and dermis near the posterior margin of the bulb of the vestibule.

### 3.4. Low-Angle Mediolateral Incision

In 75% of cadavers, the bulbospongiosus muscle, greater vestibular gland, and posterior labial nerves were in the plane of the incision. The superficial perineal nerve and its branches, as well as the bulb of the vestibule, were in the incisive plane in 25% of cadavers. The perineal body and EAS was not at risk of injury.

## 4. Discussion

The use of preserved cadavers for the purposes of informing surgical approaches has strengths and limitations. The formalin fixation during the embalming process produces cross-linking of proteoglycan monomers, making connective tissue stiffer than in living or fresh specimens. This has the benefit of making connective tissue structures opaque and easy to observe, as well as maintaining structures in situ during the dissection process to help maintain their anatomical location and relationships, which would not be the case with fresh samples or living anatomy during childbirth. Tissue biomechanical properties in embalmed cadavers are not representative of in vivo conditions, and therefore conclusions cannot be definitively drawn regarding the changes in anatomy that may be induced during surgery or the integrity of the tissues. However, clinical anatomical studies such as this one provide an important anatomical map to the body that would not otherwise be possible by detailing anatomical relationships that can otherwise be obscured or destroyed in living subjects. We, therefore, restrict our discussion of the findings to variations in anatomical location and structural relationships.

While the angle of the incision made during crowning does not remain constant after delivery due to anatomic changes in the perineum, careful dissections of the region can help the clinician make educated estimates of the structures that will be incised during episiotomy. In recent years, clinicians have debated the efficacy of routine episiotomy while working to define objective criteria to determine when episiotomy is indicated, including maternal perineal size, fetal size, and gestational timing, among other factors, and these discussions have led to several papers [10,14,29]. However, episiotomy is still sometimes diagnosed as medically necessary, especially when birth must be expedited in times of fetal distress during shoulder and other types of dystocia, which may be partly or wholly exacerbated by specifics of maternal perineal anatomy, usually in terms of small maternal perineal size [5,6]. In such instances, episiotomy may reduce occurrences of spontaneous perineal laceration, which is strongly correlated with pelvic organ prolapse and other complications later in life, when episiotomy alone is not [30]. In instances where spontaneous tearing does occur, it is most likely to happen in the midline [31]. In relation to this, spontaneous tearing is more common with midline episiotomy incisions in comparison to mediolateral incisions [32,33]. Our dissections implicate collagen fiber orientation in the connective tissue of the perineal body. These fibers course parallel to the midline incision in the sagittal plane, spanning the posterior fourchette and approaching the subcutaneous portion of the external anal sphincter. Their sagittal orientation places the path of least resistance in the midline, allowing spontaneous lacerations to result from stretching during labor as these fibers separate.

The small area of the perineum contains anatomy relevant to urinary, fecal, and sexual health, with no “safe” area for episiotomy where incision will not damage structure. However, our dissections confirmed that the perineal body was not a prominent site of muscle attachment in the majority of females we studied. This finding has been reported in other anatomical and histological investigations that describe the perineal body as having little or no insertion of striated fibers of the external anal sphincter or bulbospongiosus into the perineal body [33,34,35], and therefore it may not provide substantial protection from tearing during delivery. The expectation that the perineal body provides a major site of muscle attachments in all females selected for episiotomy may bias decisions about where to perform episiotomy, as well as surgical techniques on perineum reconstruction should tearing occur. Rather than considering the bulbospongiosus muscles and the superficial external anal sphincter as discrete, circular muscles with a common attachment to the anterior and posterior portions of the perineal body, our dissections agree with previous observations that anatomically these muscles typically comprise a single, continuous sling surrounding both the vaginal orifice and the external anal orifice that does not attach at the midline. Both portions of this sling share a common innervation from the superficial perineal nerve, which approaches the musculature laterally (Figure 3 and Figure 4) [36]. Therefore, midline episiotomy incisions would not pose a serious risk to these neuromuscular structures in the females we studied. Conversely, however, since the perineal body lacked strong muscular support in our sample of females, midline episiotomy may lead to statistically larger numbers of third- and fourth-degree tears. In addition, we found considerable variation in the distance from the posterior fourchette to the anal orifice, indicating the size of the perineal body. Incision lengths in our study were comparable to those performed in surgery [15] and consistently these at least partially bisected the perineal body. These factors are significant because the perineal body length is a large risk factor that is negatively correlated with spontaneous laceration [37,38]. Midline episiotomy in females with shorter perinea warrants extreme caution or should perhaps be avoided altogether.

In comparison, our dissections find that mediolateral episiotomy approaches do not endanger the perineal body or the subcutaneous external anal sphincter directly. Mediolateral incisions were oblique to connective tissue fiber orientation in our dissections. Our observations corroborate the results from studies of patient outcomes showing that mediolateral incisions during crowning protect against postpartum fecal incontinence from third- and fourth-degree perineal tearing by diverting forces away from the subcutaneous external anal sphincter and the perineal body [39,40,41]. However, indirect injury to the superficial external anal sphincter that is not visually evident may be diagnosed sonographically and is significant enough to elicit complaints of postpartum fecal incontinence [42,43]. We find that mediolateral incisions jeopardize the bulbospongiosus muscle, which as discussed above extends posteriorly beyond the bulb of the vestibule to be continuous with fibers of the superficial external anal sphincter to create a sling encompassing both the vaginal and external anal orifices. Anatomically, injury to these fibers may affect fecal continence by causing asymmetric contractions around the anal canal. Endoanal ultrasound studies often display this asymmetric tearing pattern [43,44]. Additionally, bulbospongiosus and the superficial external anal sphincter share innervation [34,45]. The functionality of the external anal sphincter may be impaired if nerve damage, either through incision or traction, occurs. The risk of nerve damage is greater in mediolateral incision because the perineal nerve and its branches course lateral to midline, while midline episiotomy does not anatomically endanger the superficial or deep EAS. Even if the subcutaneous EAS is injured in midline incision, this gracile muscle will heal with its nerve supply intact, and therefore will not affect long-term fecal continence. These anatomic findings should be considered when contemplating episiotomy approaches.

We also find that mediolateral incisions place several important structures related to sexual health at greater risk for injury in comparison to the midline incision, including the bulbs of the vestibule, bulbospongiosus muscle, vestibular gland, branches of the deep perineal nerve, and the trunk of the superficial perineal nerve, along with its terminal branches, the posterior labial nerves. The incisive plane endangered these structures, which serve important sexual functions, in one or more of the mediolateral incision angles. The skin in this region is well innervated and contains more mechanosensory Merkel’s cells than any other epithelium in the body [46]. The bulbs of the vestibule are the physiological seat of female orgasm, and they, along with the labia minora, swell during arousal and orgasm [46,47,48]. Greater vestibular glands are the source of preorgasmic vaginal secretions that contribute to lubrication and protection of the vagina during intercourse [49]. The bulbospongiosus–superficial external anal sphincter muscular complex contracts during sexual arousal to maintain blood within the bulbs of the vestibule, which contributes to clitoral erection, as well as providing the contractions of orgasm [34,35]. Women with midline episiotomies report both a shorter time before they engage in sexual activity than those that receive mediolateral episiotomies, and they also report no change in postpartum orgasm number; it may be that direct or indirect injury to these sexual structures is implicated in changes in postpartum sexual behavior [50,51]. Discussions of sexual activity have historically been avoided in postpartum examinations, leading to the suspected underreporting of sexual dysfunction [52,53]. Additional study is needed to evaluate the effects of episiotomy on sexual anatomy and function.

Recovery time from injury to perineal anatomy is another factor that should be considered when episiotomy is indicated. However, most surveys investigating parturition outcomes collect data relatively soon after birth, and long-term functional deficits may fail to be noted. One study that followed up with mothers at three and six months postpartum found that fecal incontinence was elevated three months after perineal tearing following midline episiotomy, but by six months the differences in fecal continence were no longer statistically significant [54]. A 10-year follow-up study showed that rates of fecal incontinence were similar between those who had perineal tearing during delivery and those who did not, regardless of whether episiotomy was performed [55]. These data seem to suggest the episiotomy type may not differ in their long-term consequences, but it is unclear if additional data from larger studies would confirm this same pattern. It is also unclear how other complications, such as urinary incontinence, sexual dysfunction, and pelvic organ prolapse, differ in postpartum years with episiotomy type, as sufficient data are lacking. Nonetheless, clinicians should incorporate the risk of short- and long-term injury into the decision to perform midline versus mediolateral episiotomy accordingly. Similarly, knowledge of perineal anatomy should be applied when diagnosing postepisiotomy complications.

## 5. Conclusions

We find that midline and mediolateral episiotomy incisions each pose unique risks to perineal anatomy, and there is no incision site that does not endanger structure. A better understanding of these risks and of the relevant anatomy is important, as clinician knowledge of perineal anatomy has been reported to be “suboptimal” and may affect individual approaches to reducing the risk of obstetric perineal injury [56]. In fact, most clinicians perform the type of episiotomy they learned in postgraduate training [21], which in the United States is the midline approach and in Europe is the mediolateral approach [26,27,57], rather than tailoring their incision to the unique circumstance of their patient. Risk to perineal anatomy should be part of the decision-making process, as should short- and long-term risks to fecal continence and sexual health. Clinician discretion is needed when balancing the risks to maternal perineal anatomy during vaginal delivery when considering episiotomy. A complete knowledge of perineal anatomy also aids the diagnosis of OASIS and other complications related to episiotomy incisions.

## Figures and Tables

**Figure 1 diagnostics-11-00221-f001:**
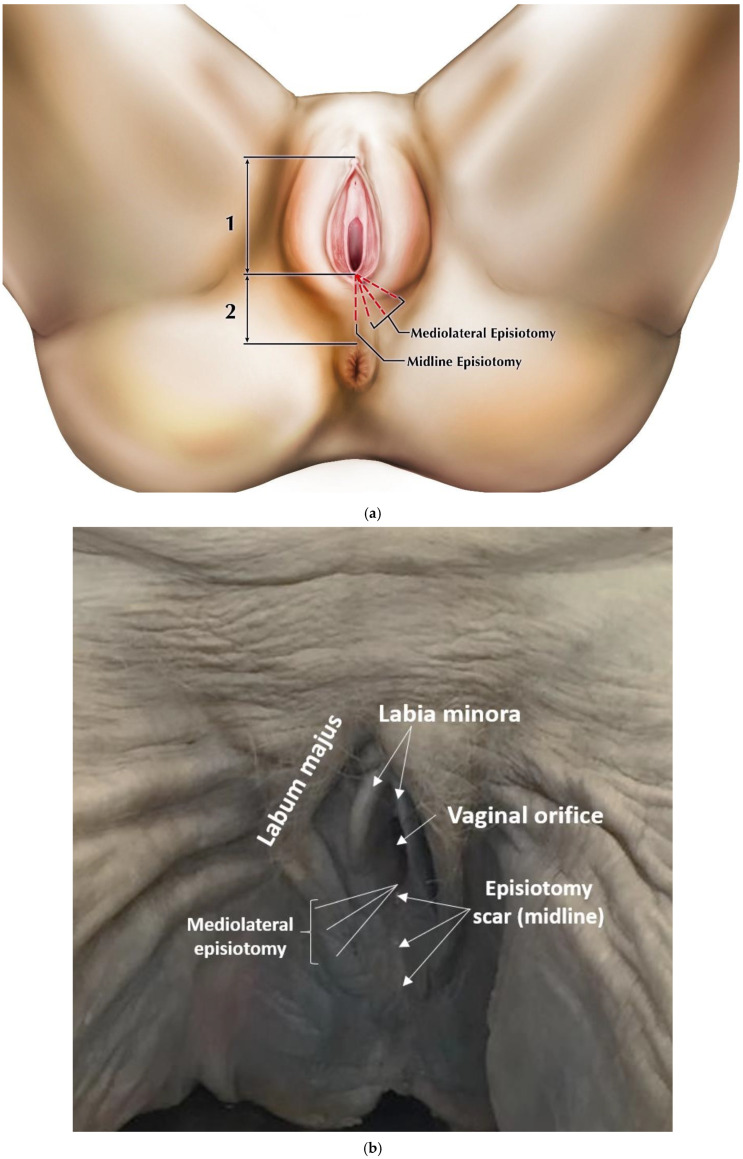
(**a**)**.** Origins and angles of incisions. Incisions were made in the midline and mediolaterally at high, medium, and low angles. (**b**). Undissected female perineum, with mediolateral episiotomy incisions indicated. This specimen exhibited an already existing episiotomy scar from a midline incision. All incisions originated at the vaginal posterior fourchette.

**Figure 2 diagnostics-11-00221-f002:**
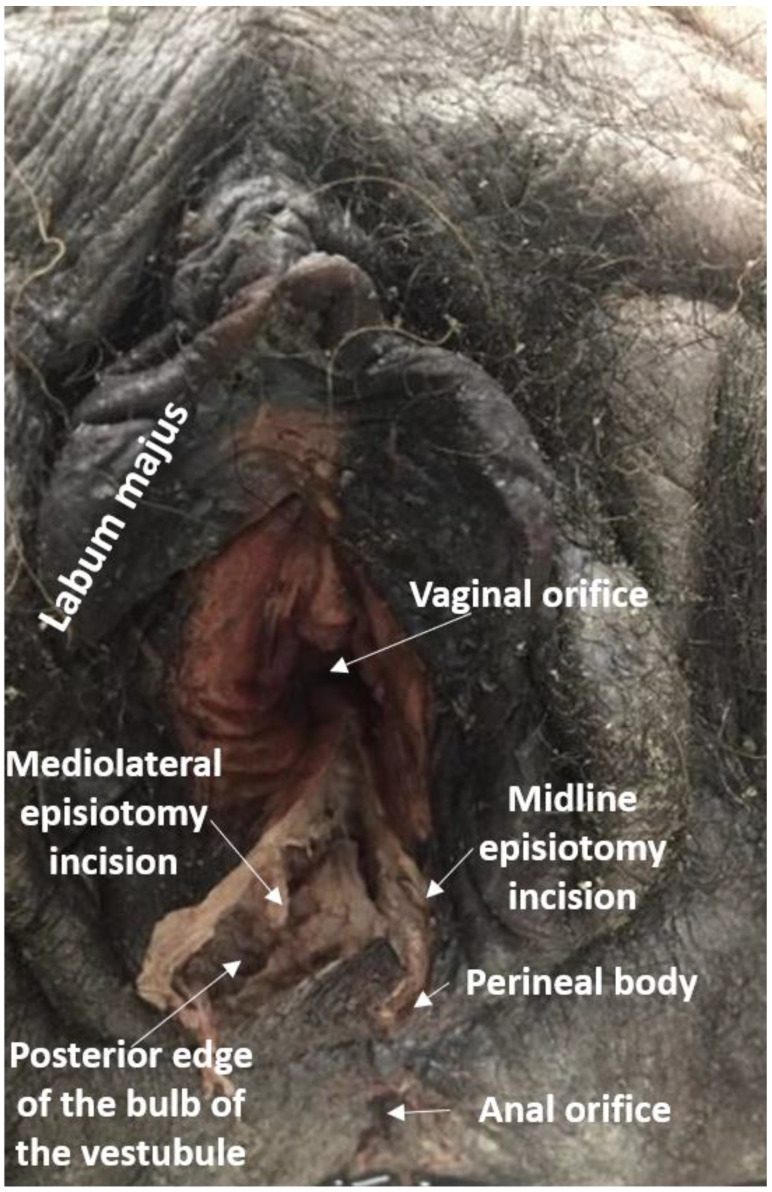
Female perineum with a midline episiotomy incision and a low-angle mediolateral incision. Midline episiotomies endanger the perineal body and many mediolateral incisions endanger the bulb of the vestibule and associated neurovasculature, as depicted here.

**Figure 3 diagnostics-11-00221-f003:**
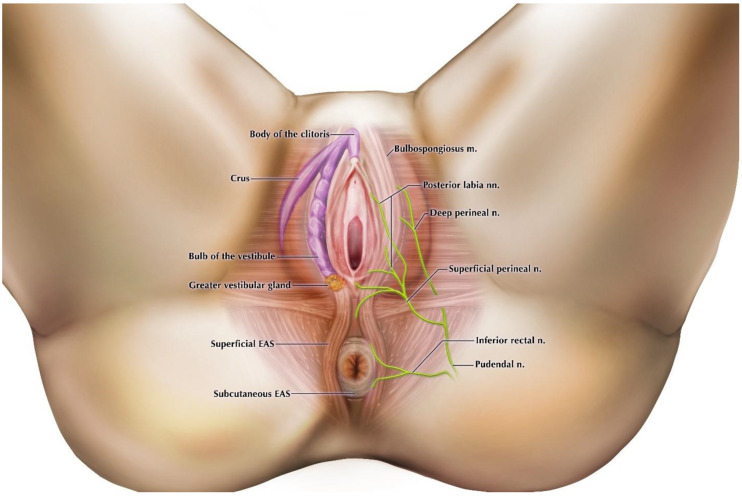
Representative anatomy of the perineum based on our dissections. Most commonly, the perineal body was not a major site of muscle attachment. The bulbospongiosus muscle was continuous with the superficial portion of the external anal sphincter (EAS), both innervated by branches of the perineal nerves. The inferior rectal nerve innervated the subcutaneous EAS and skin around the anus. The bulb of the vestibule extended to the posterior fourchette, with the greater vestibular gland anchored to its posterior margin.

**Figure 4 diagnostics-11-00221-f004:**
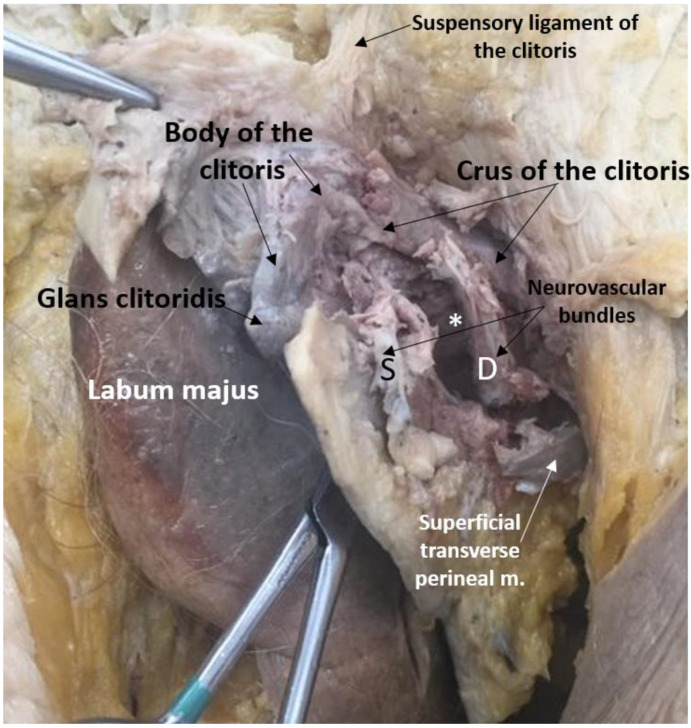
Female perineum dissected to reveal all perineal structures accessed by continuing a mediolateral episiotomy incision. The neurovascular bundles contain branches of the superficial (S) and deep (D) perineal nerves and the internal pudendal artery. *Bulb of the vestibule overlain by the bulbospongiosus muscle.

**Figure 5 diagnostics-11-00221-f005:**
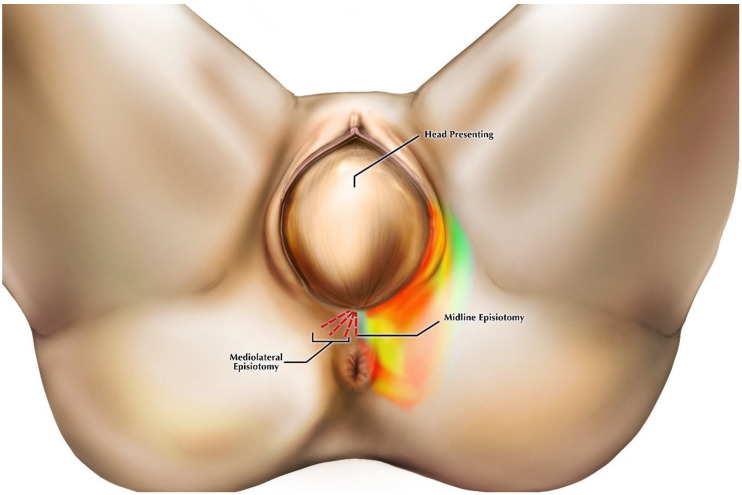
Heat map of risk of perineal structure injury at the time of crowning based on anatomic variation in our sample. Red-orange indicates high risk and green-blue low risk. Red dashed lines show the locations of episiotomy incisions.

**Table 1 diagnostics-11-00221-t001:** Percentage of females with structures located in the plane of incision.

Structures in the Incisive Plane	Midline Incision *N* = 31	Mediolateral Incision *N* = 30		
	Angle: 0°	Low Angle: 10–15° *N* = 9	Medium Angle: 16–44° *N* = 10	High Angle: ≥45° *N* = 11
Perineal body	14.3%	0.0%	0.0%	0.0%
Fibers of external anal sphincter	16.6%	0.0%	0.0%	0.0%
Bulbospongiosus	0.0%	75.0%	80.0%	100%
Bulb of the vestibule	0.0%	75.0%	80.0%	100%
Deep perineal nerve branches	0.0%	25.0%	100%	40.0%
Greater vestibular gland	0.0%	75.0%	40.0%	0.0%

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
