# Peer review of "Midline and Mediolateral Episiotomy: Risk Assessment Based on Clinical Anatomy"

_diagnostics, 2021, doi:10.3390/diagnostics11020221_

Round 1

Reviewer 1 Report

This is an interesting manuscript about Midline and Mediolateral Episiotomy: Risk Assessment Based on Clinical
Anatomy. This manuscript need some major revisions in order to be published.

At Material and method, the second paragraph have to be moved at the discussion.

About episiotomy we need few more details. How did they perform it ? What was the  dimensions of that? How did they apply the internal vaginal tension in order to perform the episiotomy? 

It will be useful to have some cadaveric pictures of the  1) episiotomy 2) measurements 3) dissection and 4) the findings through the incision. Also, figures 1 and 3 could be replaced with cadaveric pictures.  

There was 47 cadaveric specimens but 31 midline incisions were performed and it the 100% of the specimens which mean that something is wrong. Also it is mentioned at abstract that 94 incisions performed But at the method section it is  referred that the incisions was 61. it seems to be wrong. 

it will be useful to have a table with all findings in each  type of incision.

There are many limitations in a suc h cadaveric study that it is not discussed clearly.

The conclusions are not clear. We need to know where is the more safe zone in what length  and it what depth.

Reviewer 2 Report

Very interesting manuscript.

I encourage the Authors to read the following manuscript and cite it:

    Acknowledging the use of human cadaveric tissues in research papers: Recommendations from anatomical journal editors By: Iwanaga, Joe; Singh, Vishram; Ohtsuka, Aiji; et al. CLINICAL ANATOMY   Volume: ‏ 34   Issue: ‏ 1   Pages: ‏ 2-4   Published: ‏ JAN 2021

ref. 44 - Szopińska (instead of Szopinńska).

Author Response

Thanks to this reviewer for the citation suggestion; we have cited the recommended paper and amended our Acknowledgement section.

Round 2

Reviewer 1 Report

I read the revised manuscript about Midline and Mediolateral Episiotomy: Risk Assessment Based on Clinical Anatomy. Unfortunately the authors didn't manage to address all the previous comments. As the area of the pelvic floor is very confused lt is important for a cadaveric study to be very detailed in the method of dissection and mapping of the elements of cadaveric area.  This manuscript is lacked from a dissection description of that difficult area. Any anatomist and surgeon knows that dissection of this area is very difficult and confused.  Moreover every cadaveric study must be supported from detailed cadaveric pictures and not from schematic figures (they can be used additionally). Finally the table still be inappropriate. It is not referred which of the specimens has what combinations of the incisions. For that reasons this manuscript cannot be published.

Author Response

Deepest thanks for pointing out our numbers discrepancy--after you pointed out this inconsistency I went back through the different manuscript versions we had and indeed only 61 dissections were completed. Our original intention was to perform two dissections per cadaver, but in the process of performing the study we found that midline dissections could not be performed on every cadaver that had a lateral dissection, due to perineal size and other cadaver-specific reasons that would not allow two discrete dissections with independent results. This information has been corrected and added to the table.

We have now added three figures based on cadaveric photos, Figure 1b, Figure 2, and Figure 4. The original line drawing figures have been retained as Figure1a, Figure 3, and Figure 5.

Round 3

Reviewer 1 Report

I read this manuscript after re-revision that managed to addressed all the suggestion and I believe that it can be published in this version.